# High-Risk Clone of *Klebsiella pneumoniae* Co-Harbouring Class A and D Carbapenemases in Italy

**DOI:** 10.3390/ijerph19052623

**Published:** 2022-02-24

**Authors:** Arcadia Del Rio, Narcisa Muresu, Giovanni Sotgiu, Laura Saderi, Illari Sechi, Andrea Cossu, Manuela Usai, Alessandra Palmieri, Bianca Maria Are, Giovanna Deiana, Clementina Cocuzza, Marianna Martinelli, Enrico Calaresu, Andrea Fausto Piana

**Affiliations:** 1Biomedical Science PhD School, Biomedical Science Department, University of Sassari, 07100 Sassari, Italy; delrio.arcadia2@gmail.com (A.D.R.); giovanna.deiana90@gmail.com (G.D.); 2Hygiene Unit, Department of Medical, Surgical and Experimental Sciences, University of Sassari, 07100 Sassari, Italy; narcisamuresu@outlook.com (N.M.); illasechi@uniss.it (I.S.); andreacossu@uniss.it (A.C.); luca@uniss.it (A.P.); bianca.are@aousassari.it (B.M.A.); piana@uniss.it (A.F.P.); 3Clinical Epidemiology and Medical Statistics Unit, Department of Medical, Surgical and Experimental Sciences, University of Sassari, 07100 Sassari, Italy; lsaderi@uniss.it; 4Department of Humanistic and Social Sciences, University of Sassari, 07100 Sassari, Italy; manuelausai@hotmail.com; 5Department of Medicine and Surgery, University of Milano-Bicocca, 20900 Monza, Italy; clementina.cocuzza@unimib.it (C.C.); marianna.martinelli@unimib.it (M.M.); e.calaresu1@gmail.com (E.C.)

**Keywords:** antimicrobial resistance, Carbapenemase genes, *Klebsiella pneumoniae*, nosocomial infections

## Abstract

*Background:* Carbapenem-resistant *Klebsiella pneumoniae* (CR-Kp) is endemic globally, causing severe infections in hospitalized patients. Surveillance programs help monitor and promptly identify the emergence of new clones. We reported the rapid spread of a novel clone of *K. pneumoniae* co-harbouring class A and D carbapenemases in colonized patients, and the potential risk factors involved in the development of infections. *Methods:* Rectal swabs were used for microbiological analyses and detection of the most common carbapenemase encoding genes by real-time PCR (i.e., blaKPC, blaOXA-48, blaNDM, blaVIM, and blaIMP). All strains co-harbouring KPC and OXA-48 genes were evaluated. For each patient, the following variables were collected: age, sex, length and ward of stay, device use, and outcome. Clonality of CR-Kp was assessed by preliminary pulsed field gel electrophoresis (PFGE), followed by multi-locus sequence typing (MLST) analyses. *Results:* A total of 127 isolates of *K. pneumoniae* co-harbouring KPC and OXA-48 were collected between September 2019 and December 2020. The median age (IQR) of patients was 70 (61–77). More than 40% of patients were admitted to intensive care unit (ICU). Around 25% of patients developed an invasive infection, the majority of which were respiratory tract infections (17/31; 54.8%). ICU stay and invasive infection increased the risk of mortality (OR: 5.39, 95% CI: 2.42–12.00; OR 6.12, 95% CI: 2.55–14.69, respectively; *p*-value ≤ 0.001). The antibiotic susceptibility test showed a resistance profile for almost all antibiotics considered. Monoclonal origin was confirmed by PFGE and MLST showing a similar restriction pattern and belonging to ST-512. *Conclusions:* We report the spread and the marked antibiotic resistance profiles of *K. pneumoniae* strains co-producing KPC and OXA-48. Further study could clarify the roles of clinical and microbiological variables in the development of invasive infection and increasing risk of mortality, in colonized patients.

## 1. Introduction

Hospital-acquired infections (HAIs) are associated with significant morbidity and mortality [1]. More than 4 million cases of HAIs occur annually in Europe, with ~37,000 deaths [2], and >60% of them are caused by multi-drug resistant (MDR) microorganisms, such as methicillin-resistant *Staphylococcus aureus* (MRSA), extended-spectrum beta-lactamase (ESBL)-producing bacteria, carbapenem-resistant *Enterobacteriaceae* (CRE), carbapenem-resistant *Acinetobacter baumannii*, and MDR *Pseudomonas aeruginosa* [2,3]. 

The high MDR rate, especially in nosocomial settings, can reduce the availability of effective therapeutic options, increasing mortality and healthcare costs. Mortality associated with CRE ranges from 30% to 70% and exceeds 50% in case of bloodstream infections. Carbapenems act as inhibitors in the synthesis of the bacterial cell wall, binding to penicillin-binding proteins. In the past, carbapenems were often recognized as “last-line agents” in the treatment of infections caused by Gram-negative and Gram-positive bacteria, based on their broad spectrum of activity. Among the acquired mechanisms of resistance, the inactivation of antimicrobial drugs following the production of carbapenemases is one of the most frequent mechanisms of resistance [4,5]. Carbapenem-resistant *K. pneumoniae* (CR-Kp) is endemic in USA, Brazil, Argentina, Colombia, China, Taiwan, and Europe. Class A carbapenemases target the most prevalent β-lactamases (*Klebsiella pneumoniae* carbapenemases, KPCs) found in CR-Kp isolates. Class B metallo β-lactamases are encoded by VIM (Verona integron-encoded metallo-β-lactamase), IMP (Imipenemase metallo-β-lactamase), and NDM (New-Delhi metallo-β-lactamase) genes. Class D genes include OXA-48 carbapenemases, which show high activity against penicillins, low activity against carbapenems, and weak activity against broad-spectrum cephalosporins [6]. 

CRE are frequently transmitted through healthcare personnel’s hands or contaminated medical equipment or surfaces [7].

Intestinal colonization by CRE can be a risk factor for infections (respiratory tract, urinary tract, bloodstream, surgical wounds, skin, and soft tissue) in hospitalized patients, with an associated mortality rate ranging from 30% to 75% [5].

Identification of carriers and adoption of containment measures are key for the prevention of further colonization cases and clinical outbreaks through the implementation of surveillance programs [8].

After the first detection of KPC-producing *K. pneumoniae* in 2015 at the University Hospital of Sassari, Italy [9], a surveillance program was implemented: rectal swabs upon hospital admission and screening for close contacts of index cases were systematically carried out [10]. In the present study we describe the epidemiological characteristics of a novel clone of *K. pneumoniae* co-harbouring KPC and OXA-48 carbapenemases isolated from colonized patients, and the associations between demographic/clinical variables and invasive infection and death. The findings could help plan appropriate infection control measures and preventive interventions.

## 2. Materials and Methods

An observational retrospective study was carried out at the University of Sassari, Italy. We considered all strains of *K. pneumoniae* co-producing KPC and OXA-48, since first detection in September 2019 to December 2020, isolated during the activities of the screening program. Only the first positive rectal swabs, collected during the activities of the screening program, were selected for the present study. 

Colonization status was defined as a detection of CRE from rectal swab or faeces without any evidence of active infection, whereas infection, refers to the detection of carbapenem-resistant K. *pneumoniae* in patients with infection-related symptoms, isolated from clinical specimens or a normally sterile site (i.e., blood or bronchoalveolar lavage, respectively).

The following variables were retrospectively collected: demographics (sex, age), comorbidities, history of community or healthcare-associated infections, ward of admission, length of hospital stay, use of devices, and final clinical outcome.

### 2.1. Microbiological Analysis

Rectal swabs were plated into Chromid Carba Smart Agar (bioMérieux, Italia) [11]; after overnight incubation, identification was performed using the Vitek-2 System (bioMerieux, Marcy l’Etoile, France) [12]. Carbapenemase encoding genes, blaKPC, blaNDM, blaVIM, blaOXA-48, and blaIMP were detected by real time-PCR using the commercial kit Allplex Entero-DR assay. Antimicrobial resistance profile and clonality of CR-Kp strains were assessed based on a representative set of isolates, selected by ward of admission and time of detection. Drug susceptibility test and minimum inhibitory concentration (MIC) values were assessed referring to breakpoints of the European Committee on Antimicrobial Susceptibility Testing (EUCAST) [13]. Clonality was preliminarily investigated by pulsed-field gel electrophoresis (PFGE) and interpreted according to Tenover criteria [14]. Bacteria were suspended in agarose disk before DNA extraction and purification. Bacterial genomes were cut by ApaI restriction enzyme, and single fragments were separated by agarose PFGE using a Clamped Homogeneous Electric Fields DRII System (BIORAD). Images of gels were captured by Image Master Program, and positions of electrophoretic bands and the phylogenetic dendrograms were obtained by GelCompar II (Applied Math) (Appendix A) [9]. Isolates were clonally related if the Dice coefficient was >80%, whereas patterns with indistinguishable PFGE banding patterns (similarity coefficient > 97%) belonged to the same subtype. Multi-locus sequence typing (MLST) analysis was performed to measure genetic relatedness and sequence variation between alleles, by amplifying and sequencing the internal fragments of seven *K. pneumoniae* housekeeping genes: gapA, infB, mdh, pgi, phoE, rpoB, and tonB (Appendix A: MLST analyses). Sequence analysis was carried out using the software Bioedit, and each single locus was compared with those included in the database of the Pasteur Institute to evaluate percentage similarity and compatibility [15].

### 2.2. Statistical Analysis

The following variables were collected on an electronic form: demographic (i.e., sex, age), clinical (i.e., hospital and length of stay, use of device, comorbidities, outcome), and microbiological (i.e., occurrence of invasive infection, antimicrobial susceptibility testing, MLST, molecular analyses for carbapenemase detection) variables. Absolute and relative (%) frequencies, means and standard deviations (SD), or medians and interquartile ranges (IQR) were used to summarize qualitative and normally and non-normally distributed quantitative variables, respectively. Data were analysed using STATA Version 17 (StataCorp, College Station, TX, USA).

## 3. Results

A total of 127 isolates of *Klebsiella pneumoniae* co-harbouring KPC and OXA-48 were detected between September 2019 and December 2020 at the University Hospital of Sassari, Italy. Fifty-three (41.7%) patients were female, and the median (IQR) age was 70 (61–77) years (Table 1). 

Patients were frequently admitted to the ICU (51/127; 40.2%), surgical wards (38/127; 29.9%), internal medicine wards (31/127; 24.4%), long-term care wards (4/127; 3.2%), and the infectious diseases ward (3/127; 2.4%) (Figure 1). Median (IQR) length of hospital stay was 46 (29–68) days.

Over 80% (103/127; 81.1%) of patients had at least one clinical device: central venous catheter (68/127, 66.0%), mechanical ventilation (51/127, 49.5%), arterial line (37/127, 35.9%), and tracheal tube (31/127, 30.1%). The median (IQR) number of devices per patient was two (1–3).

The median (IQR) number of days between hospital admission and first detection of *K. pneumoniae* KPC-OXA-48 positive swab was 26 (13–43) days.

More than half the patients (84/127, 66.1%) had at least one comorbidity, including hypertension (37, 44.1%), cancer (22, 26.2%), diabetes (21, 25.0%), and cardiomyopathies (20, 23.8%). Three quarters (63/84; 75.0%) of patients had two or more comorbidities (Table 2). 

An invasive infection caused by KPC-OXA-48 *K. pneumoniae* occurred in 31/127 (24.4%) patients; respiratory and bloodstream infections occurred in 17/31 (54.8%) and 14/31 (45.2%) patients, respectively. The median (IQR) number of days between a diagnosis of colonization and development of an invasive infection was 10 (0–22) days.

A total of 42 (33.1%) deaths were reported. ICU stay was associated with an increased risk of mortality (OR: 5.39, 95% CI: 2.42–12.00; *p*-value ≤ 0.001) and an increased risk of invasive infection (OR 6.12, 95% CI: 2.55–14.69; *p*-value < 0.001) (Table 3).

Logistic regression analysis was carried out to assess the roles of demographic, clinical, and epidemiological factors in the occurrence of infection. It showed that diabetic patients (OR: 3.26, 95% CI: 1.08–990; *p*-value: 0.04) and ICU stay (OR: 3.81, 95% CI: 1.63–8.93; *p*-value: 0.002) were significantly associated with infection (Table 4).

The antibiotic susceptibility test, performed for 67 (52.8%) isolates, showed a complicated drug resistance pattern to almost all classes of antibiotics, including carbapenems, cephalosporins, penicillins, and aminoglycosides (Table 5).

A total of 44 representative samples of KPC and OXA-48 co-producing *Klebsiella pneumoniae* strains were analysed with PFGE. The PFGE analysis showed a similar restriction pattern; all strains were related to the same subtype (Dice coefficient > 80%; similarity coefficient > 97%). The MLST analysis showed a monoclonal origin of isolates, which belonged to the same sequence type, i.e., ST-512.

## 4. Discussion

The present study describes phenotypic and molecular characteristics of *K. pneumoniae* co-harbouring KPC and OXA-48, isolated at the university hospital of Sassari, Italy. A patient colonized by *K. pneumoniae* co-producing KPC and OXA-48 and transferred from a hospital located in northern Italy to the ICU of the university hospital of Sassari was the index case: the patient was immediately associated with a new case of colonization in a close contact and with rapid spread in the hospital wards. To the best of our knowledge, this is the first report of *K. pneumoniae* co-harbouring KPC and OXA-48 carbapenemases in Italy.

The incidence of CPE (carbapenemase producing *Enterobacteriaceae*) in Italy is high: almost one third of *Klebsiella pneumoniae* strains isolated from invasive infections are carbapenem-resistant, and the most common gene of resistance is KPC [16,17]. The KPC family includes more than 40 variants characterized by activity against cephalosporins, monobactams, and carbapenems [18]. Moreover, although the Italian prevalence is only 0.5% [17], class D carbapenemase OXA-48 is the most prevalent carbapenemase in Enterobacterales isolated in North Africa and in the Mediterranean area [19]. Class D carbapenemases showed high activity against aztreonam, extended spectrum cephalosporins, and carbapenems, especially when combined with other resistance mechanisms (i.e., reduced permeability of membrane, and co-production of ESBL encoding genes or AmpC enzymes) [19].

Several reports show the worldwide emergence of strains with double or multiple carbapenemases, reducing the available treatment options, mostly in geographical areas where the endemicity of carbapenemase-producing Enterobacteriaceae is relevant [20]. Recently, Chen and Colleagues described the rapid spread of *Klebsiella pneumoniae* ST11 co-expressing class A and D carbapenemases in a teaching hospital in Taiwan [21]. The majority of carbapenemases genes can be transferred horizontally by mobile genetic elements, facilitating the evolutionary success of multiple-carbapenemase-producing strains, mainly in hospital settings where continuous antimicrobial resistance pressure occurs [22].

Antibiotic susceptibility tests performed in our setting showed a complicated drug resistance pattern. If compared with previous data on KPC-producing *K. pneumoniae* strains [9], the percentages of resistance to aminoglycosides and to tigecycline raised from 28% to 94.2% and from 50% to 87.5%, respectively. This finding could potentially be explained by the co-expression of class A and D carbapenemases or by the presence of other additional genes localized in plasmids or mobile genetic elements [22]. Meletis and colleagues assumed that not all carbapenemases effectively hydrolyse the available beta-lactams; however, a combination of double or multiple carbapenemases causes an enhanced effect, favouring total resistance [20]. Moreover, resistance to ceftazidime/avibactam (CAZ/AVI) was observed in one third of the strains, raising concerns about the spread of emerging clones resistant to new therapeutic options. CAZ/AVI is active against class A, C, and D carbapenemases; and it is recommended for complicated urinary tract and intra-abdominal infections, and hospital-acquired and ventilator associated pneumonia caused by Gram-negative bacteria, including CR-Kp [23]. Haidar et al. [24] showed that mutations in KPC-3 variants could explain the higher MICs of CAZ/AVI and carbapenems. On this basis, the emergence of CAZ/AVI-resistant strains could be related to the presence of KPC mutants. However, further investigations could clarify the role of mutations behind the emergence of CAZ/AVI resistance.

We found an association between mortality and ICU stay, in agreement with the scientific literature focused on the roles played by some variables described in ICU wards (e.g., use of devices and prolonged antimicrobial therapies), as well-established risk factors for acquisition of CRE and higher mortality [25,26]. Moreover, higher risks of mortality and onset of infection in patients hospitalized in surgical units were associated with invasive and surgical procedures [27]. Although the mechanisms involved in the progression from colonization to infection are unclear, gastrointestinal carriage of *K. pneumoniae* is a well-established risk factor for invasive infections [28,29], as we observed in >20% of colonized patients. The small samples size could have affected the identification of statistically significant associations between demographic variables (i.e., age and gender), several comorbidities, and outcomes. However, we found significant associations between diabetes and the onset of invasive infections. Several reports highlighted the role of underlying diseases in the progression from colonization to infection: they can weaken the immune response and increasing host susceptibility to infection [28]. Further studies are needed to explain how progression occurs and potential protective factors. 

Unfortunately, missing data on previous or ongoing antimicrobial therapies did not allow us to evaluate the role of antibiotics in the onset of colonization or infections. However, interventions of antimicrobial stewardship could reduce the use of antimicrobial drugs, and then prevent the emergence of new antimicrobial resistant strains [30]. 

The mortality in our cohort was 33.1%, increasing to 50% in the ICU, in agreement with previous reports which reported a mortality rate >50% because of severe disease and poor therapeutic options [28].

A greater length of stay was associated with patients who developed invasive infections compared to colonized patients (46 VS. 71 days, respectively). Prolonged hospital stays are well-known risk factors for HAI, severe disease, death, and increased healthcare costs [25]. European guidelines strongly recommend screening swabs at hospital admission, and weekly surveillance for patients at high risk of carriage of MDR bacteria; moreover, screening of case-contacts is a key tool to contain the spread of MDR infections [31]. Early detection of carbapenem resistant *K. pneumoniae,* implementation of control measures, and prompt isolation of colonized patients may considerably reduce morbidity and mortality rates.

PFGE and MLST confirmed monoclonal circulation of ST-512, supporting national prevalence data [9,32] and explaining the hospital spread of *K. pneumoniae* co-harbouring KPC and OXA-48 strains. Moreover, the rapid global transmission of ST-258, which is different when compared with ST-512 by only one mutation in gap locus, highlights the roles of genetic characteristics which can increase pathogenicity and transmissibility [33]. 

The limitations of the present study are mainly related to the absence of some important clinical data, and its retrospective and mono-centre epidemiological nature. In addition, the small sample size limited the power of the statistical analysis, including the identification of the potential roles played by risk factors. Advanced techniques, such as plasmid analysis and whole genome sequencing, could be useful for describing epidemiological dynamics, although these are not yet recommended in the available guidelines for screening programs [31]. 

The results of our screening program helped define the main epidemiological and clinical characteristics of circulating strains, although future multi-centre studies should investigate the roles of clinical and epidemiological factors in the prognosis of colonized patients. Of note, the new clone of *K. pneumoniae* co-harbouring class A and D carbapenemases detected in our setting highlight the risk of new multi-drug resistant strains.

## 5. Conclusions

In conclusion, surveillance programs, reinforcement of contact precautions, and education of staff can reduce the impact of antimicrobial resistance globally. Moreover, molecular characterization of antimicrobial resistance determinants, such as carbapenemase encoding genes, and adoption of alternative “non-antibiotic” strategies are essential to tracking the spread of high-risk strains, to promptly providing appropriate tailored therapeutic options, and to meeting “one-health” goals.

## Figures and Tables

**Figure 1 ijerph-19-02623-f001:**
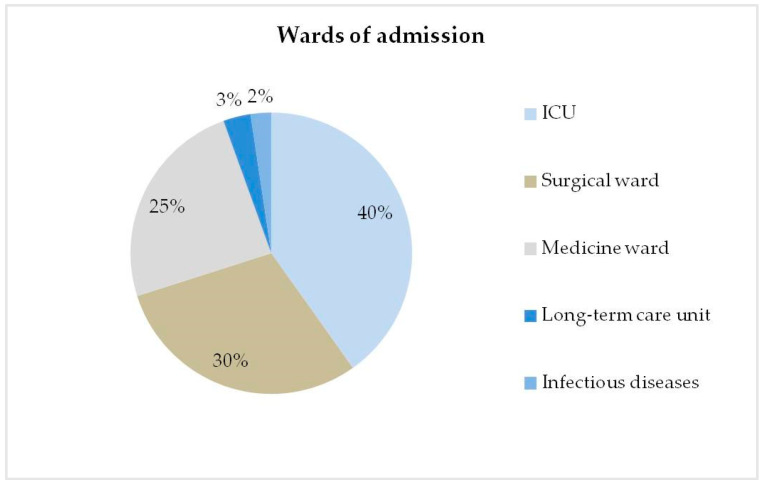
Representation of the distribution of wards of admission during the study period.

**Table 1 ijerph-19-02623-t001:** Demographic characteristics of patients involved in the study (*n* = 127).

Variables		
Female, *n* (%)		53 (41.7)
Median (IQR) age, years		70 (61–77)
Ward of admission, *n* (%)	Surgical ward	38 (29.9)
	ICU	51 (40.2)
	Medicine ward	31 (24.4)
	Infectious diseases	3 (2.4)
	Long-term care	4 (3.2)
Median (IQR) length of stay, days		46 (29–68)
Device use, *n* (%)		103 (81.1)
Median (IQR) No. devices		2 (1–3)
Device, *n* (%)	CVC	68 (66.0)
	Peripheral venous catheters	4 (3.9)
	Urinary catheters	7 (6.8)
	PEG	9 (8.7)
	Arterial line	37 (35.9)
	VAP	51 (49.5)
	Tracheal tube	31 (30.1)
	Other stomia	15 (14.5)
Median (IQR) number of days between admission and colonization		26 (13–43)
Subjects with comorbidity, *n* (%)		84 (66.1)
Invasive infections, *n* (%)	Bloodstream infections	14/31 (45.2)
	Respiratory tract infections	17/31 (54.8)
Median (IQR) number of days between diagnosis of colonization and invasive infection		10 (0–22)
Deaths, *n* (%)		42 (33.1)

**Table 2 ijerph-19-02623-t002:** Prevalence of comorbidities.

Comorbidity	Total Cohort *n*(%) (84; 100%)	One Comorbidity (21/84; 25%)	≥2 Comorbidities (63/84; 75%)
Hypertension	37 (44.1)	2 (9.5)	35 (55.6)
Cancer	22 (26.2)	9 (42.9)	13 (20.6)
Diabetes	21 (25.0)	2 (9.5)	19 (30.2)
Cardiomyopathy	20 (23.8)	1 (4.8)	19 (30.2)
Atrial fibrillation	14 (16.7)	0 (0.0)	14 (22.2)
Anaemia	14 (16.7)	2 (9.5)	12 (19.1)
Obesity	13 (15.5)	0 (0.0)	13 (20.6)
COPD	12 (14.3)	0 (0.0)	12 (19.1)
Chronic renal failure	12 (14.3)	1 (4.8)	11 (17.5)
Respiratory insufficiency	10 (11.9)	1 (4.8)	9 (14.3)
Benign prostatic hyperplasia	7 (8.3)	1 (4.8)	6 (9.5)
Asthma	4 (4.8)	1 (4.8)	3 (4.8)
Auto-immune diseases	4 (4.8)	0 (0.0)	4 (6.4)
Rheumatoid arthritis	3 (3.6)	0 (0.0)	3 (4.8)
Multiple sclerosis	3 (3.6)	1 (4.8)	2 (3.2)
Leukaemia/Myeloma	2 (2.4)	0 (0.0)	2 (3.2)

**Table 3 ijerph-19-02623-t003:** Logistic regression analysis to assess the relationships between demographic, epidemiological and clinical variables and mortality.

Variables		OR (95% CI)	*p*-Value
Female, *n* (%)		0.80 (0.38–1.70)	0.56
Median (IQR) age, years		1.03 (1.00–1.06)	0.07
Ward of admission, *n* (%)	Surgical ward	0.16 (0.05–0.48)	0.001
	ICU	5.39 (2.42–12.00)	<0.001
	Medicine	0.78 (0.32–1.89)	0.58
	Infectious diseases	/	/
	Long-term care	0.67 (0.07–6.61)	0.73
Median (IQR) length of stay, days		1.01 (1.00–1.01)	0.24
Device use		2.13 (0.73–6.17)	0.16
No. Devices		1.13 (0.72–1.78)	0.60
Median (IQR) number of days between admission and positivity		1.00 (0.99–1.02)	0.64
Subject with comorbidity, *n* (%)		0.88 (0.41–1.92)	0.76
Comorbidity, *n* (%)	Hypertension	1.28 (0.51–3.20)	0.60
	Cardiomyopathy	0.64 (0.20–1.98)	0.44
	Diabetes	1.88 (0.67–5.21)	0.23
	COPD	1.07 (0.29–3.90)	0.92
	Obesity	0.93 (0.26–3.33)	0.91
	Asthma	/	/
	Cancer	0.98 (0.35–2.78)	0.97
	Respiratory insufficiency	/	/
	Multiple sclerosis	/	/
Median (IQR) number of days between diagnosis of colonization and invasive infection		0.99 (0.98–1.01)	0.35
Invasive infections, *n* (%)		6.12 (2.55–14.69)	<0.001

**Table 4 ijerph-19-02623-t004:** Logistic regression analysis to assess the relationships between demographic, epidemiological and clinical variables and development of invasive infections.

Variables		OR (95% CI)	*p*-Value
Female, *n* (%)		0.48 (0.20–1.16)	0.10
Median (IQR) age, years		0.98 (0.96–1.01)	0.22
Ward of admission, *n* (%)	Surgical ward	0.27 (0.09–0.84)	0.02
	ICU	3.81 (1.63–8.93)	0.002
	Medicine	0.88 (0.34–2.29)	0.79
	Infectious diseases	/	/
	Long-term care	/	/
Median (IQR) length of stay, days		1.01 (1.00–1.02)	0.05
Device use		4.31 (0.95–19.51)	0.06
No. Devices		1.41 (0.89–2.31)	0.18
Median (IQR) number of days between admission and positivity		1.00 (0.99–1.01)	0.93
Subject with comorbidity, *n* (%)		0.63 (0.27–1.45)	0.28
Comorbidity, *n* (%)	Hypertension	1.81 (0.63–5.17)	0.27
	Cardiomyopathy	0.58 (0.15–2.23)	0.43
	Diabetes	3.26 (1.08–9.90)	0.04
	COPD	0.70 (0.14–3.53)	0.67
	Obesity	1.12 (0.27–4.59)	0.88
	Asthma	/	/
	Cancer	0.76 (0.22–2.62)	0.67
	Respiratory insufficiency	/	/
	Multiple sclerosis	/	/

**Table 5 ijerph-19-02623-t005:** Results of drug susceptibility testing of *Klebsiella pneumoniae* co-harbouring KPC and OXA-48.

Antibiotic	Number of Isolates (%):	
	Intermediately Resistant	Resistant
Amikacin	-	69/69 (100)
Amoxicillin/Clavulanic acid	-	67/67 (100)
Cefepime	-	38/38 (100)
Cefotaxime	-	69/69 (100)
Ceftazidime	-	69/69 (100)
Ciprofloxacin	-	69/69 (100)
Ertapenem	-	48/48 (100)
Fosfomycin	-	36/36 (100)
Gentamicin	-	65/69 (94)
Imipenem	-	25/25 (100)
Meropenem	1/69 (1)	68/69 (99)
Piperacillin/Tazobactam	-	67/67 (100)
Tigecycline	4/32 (12.5)	28/32 (88)
Trimethoprim/Sulfamethoxazole	-	67/67 (100)
Ceftazidime/Avibactam	-	6/21 (29)
Ceftolozane/Tazobactam	-	21/21 (100)
Tobramycin	-	23/23 (100)

## Data Availability

The data is available in case it is requested for motivated reasons.

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
