# Peer review of "High-Risk Clone of Klebsiella pneumoniae Co-Harbouring Class A and D Carbapenemases in Italy"

_ijerph, 2022, doi:10.3390/ijerph19052623_

Round 1

Reviewer 1 Report

Although the work is of great importance, the manuscript needs a lot of revision.

first, in the introduction, the research question are not clear and the rational is mess.

The study design, study type,  and sampling size are not clear.

In methods, molecular typing is insufficient,, vitek identification is poor, mass spectrometric identification give more results.

Table 3, hospital stay information is not clear

Author Response

Reviewer #1

R1:”Although the work is of great importance, the manuscript needs a lot of revision. First, in the introduction, the research question are not clear and the rational is mess.”

AA: We thank the Reviewer for having provided an important contribution to the improvement of our manuscript. Following his/her suggestions, we performed several changes in the text.

Surveillance of infections caused by carbapenemase-producing Enterobacteriaceae is key to adequately plan and tailor preventive strategies. The present study reports on the emergence of a novel clone of K.pneumoniae characterized by a complicated antimicrobial resistance profile, which can complicate the clinical management of infected patients. In particular, the aim of our study was to assess the relationship between demographic/clinical variables and occurrence of infection and mortality in patients colonized by K.pneumoniae co-harboring KPC and OXA-48. As we stated in the paragraph on study limitations, the small sample size could affect the statistical power of the main findings. However, the detection of some potential risk factors suggests the reinforcement of infection control measures in our settings and future research on this topic.

Following the point raised by the Reviewer, the aim of the study was edited as follows: “In the present study we described the epidemiological characteristics of a novel clone of K.pneumoniae co-harbouring KPC and OXA-48 carbapenemases isolated from colonized patients, and the association between demographic/clinical variables and invasive infection and death. The findings could help plan appropriate infection control measures and preventive interventions.

R1:”The study design, study type,  and sampling size are not clear.”

AA: We thank the Reviewer for having raised those points. We conducted an observational, retrospective study aimed at describing the emergence and spread of K.pneumoniae co-producing KPC and OXA-48 strains, since the first detection in September 2019 until December 2020, in an Italian university hospital, as well at evaluating potential risk factors for invasive infection and mortality. A representative selection of biological samples, based on time of detection and ward of admission, underwent antimicrobial susceptibility testing and analysis of clonality by PFGE and MLST. Accordingly, more details were added in the manuscript (Please, see Materials and Methods section, page 2).

R1:”In methods, molecular typing is insufficient, vitek identification is poor, mass spectrometric identification give more results.”

AA: We thank the Reviewer for this important comment. Mass spectrometry represents a valid alternative to standard microorganism identification in clinical laboratories. This technique combines the ease of use with a significant reduction in time of diagnosis. However, its application for the identification of carbapenemase-producing strains is still uncertain and shows several limitations, due to the limited number of spectra that could affect the microbiological diagnosis, mainly at species level and for carbapenemase discrimination. On the other hand, VITEK2 system, designed on a growth-based technology, allowed us to identify microorganisms with an excellent accuracy (96%-99% probability) and, simultaneously, to carry out the antibiotic susceptibility testing, when required. Moreover, molecular surveillance for CPE is supported by international guidelines because of its diagnostic specificity and reliability [European Centre for Disease Prevention and Control. ECDC study protocol for genomic-based surveillance of carbapenem-resistant and/or colistin-resistant Enterobacteriaceae at the EU level. Version 2.0 Stockholm: ECDC; 2018].

R1:”Table 3, hospital stay information is not clear”

AA: We thank the Reviewer for this request of clarification. Following the suggestion, the text was edited using the line “ward of admission” (Please, see tab.1, 3 and 4).

Reviewer 2 Report

Title: High-risk clone of Klebsiella pneumoniae co-harbouring class A and D carbapenemases in Italy

The study by Del Rio et al. is focused on the rapid spread of a novel clone of K. pneumoniae co-harbouring class A and D carbapenemases, and the potential risk factors involved in the development of infections. The authors have examined samples collected from patients hospitalized between September 2019 and December 2020 to detect the most common carbapenem-resistant encoding genes. The obtained microbiological data have been correlated with demographic, epidemiological, and clinical variables. Overall, the goal of the work is interesting and appropriately conceived, however, as delineated in detail below, the manuscript needs a major revision to make the manuscript more clear and easily readable.

Title:

  • The authors should change “pnumoniae” in “pneumoniae”

Abstract:

  • The abstract appropriately summarized the work performed in the present study, but contains several acronyms not still specified. The authors should define all the terms if mentioned for the first time (for example CR-Kp, PFGE, MLST, ICU).
  • Line 31: “>40%”. At the beginning of the sentence is better to write the word instead to refer to the number (“More than 40% of the patients…”) to make reading smoother.

Introduction:

  • The introduction should be enriched with information about Carbapenems (mechanism of action of this antimicrobial, why its use is so diffuse in clinic…), and the section with the Carbapenem-resistant (CR) bacterial strains should be improved with extra details about how CR strains affect the classical antimicrobial therapies and the percentage of patients survival in the hospitals. Also, the authors should clarify the role of the carbapenemases within the mechanism of resistance.

Materials and Methods:

  • Line 77-82. The authors should re-word this part with shorter sentences. This part is confusing.

Results:

  • All the collected data are shown in tables. Although the tables are informative, the authors could use different types of graphs to plot the data (for example the cake charts might help the reader to follow the obtained results)
  • At the beginning of the sentence is better to write the word instead to refer to the number to make reading smoother (for example line 231 and many other)
  • The authors should clarify the meaning of the acronyms ICU and CPE.

Discussion

  • The significance of the obtained results is appropriately described in the discussion.

Author Response

Reviewer #2:

The study by Del Rio et al. is focused on the rapid spread of a novel clone of K. pneumoniae co-harbouring class A and D carbapenemases, and the potential risk factors involved in the development of infections. The authors have examined samples collected from patients hospitalized between September 2019 and December 2020 to detect the most common carbapenem-resistant encoding genes. The obtained microbiological data have been correlated with demographic, epidemiological, and clinical variables. Overall, the goal of the work is interesting and appropriately conceived, however, as delineated in detail below, the manuscript needs a major revision to make the manuscript more clear and easily readable.

We thank the Reviewer for having provided a crucial contribution to the improvement of our manuscript.

R2:”Title: The authors should change “pnumoniae” in “pneumoniae”

AA: We thank the Reviewer for his/her suggestion. The title was edited accordingly.

R2:” Abstract: The abstract appropriately summarized the work performed in the present study, but contains several acronyms not still specified. The authors should define all the terms if mentioned for the first time (for example CR-Kp, PFGE, MLST, ICU).

AA: We thank the Reviewer for his/her suggestion. The acronyms were clarified.

Line 20: Carbapenem-resistant Klebsiella pneumoniae (CR-Kp)

Line 28: Pulsed field gel electrophoresis (PFGE)

Line 29: Multi Locus Sequence Typing (MLST)

Line 31: Intensive Care Unit (ICU)

(see Abstract section, page 1).

R2: “Line 31: “>40%”. At the beginning of the sentence is better to write the word instead to refer to the number (“More than 40% of the patients…”) to make reading smoother.

AA: We thank the Reviewer for his/her suggestion. The text was edited accordingly (see Abstract section, page 1).

R2:” Introduction: The introduction should be enriched with information about Carbapenems (mechanism of action of this antimicrobial, why its use is so diffuse in clinic…), and the section with the Carbapenem-resistant (CR) bacterial strains should be improved with extra details about how CR strains affect the classical antimicrobial therapies and the percentage of patients survival in the hospitals. Also, the authors should clarify the role of the carbapenemases within the mechanism of resistance.”

AA: We thank the Reviewer for having requested this clarification and the inclusion of further details. The following paragraph was added in the text: “The high MDR rate, especially in nosocomial settings, can reduce the availability of effective therapeutic options, increasing mortality and healthcare costs. Mortality associated to CRE ranges from 30% to 70% and exceeds 50% in case of bloodstream infections. Carbapenems act as inhibitors in the synthesis of the bacterial cell wall, binding to penicillin-binding proteins. In the past carbapenems were often recognized as “last-line agents” in the treatment of infections caused by Gram-negative and -positive, based on their broad spectrum of activity. Among the acquired mechanisms of resistance, the inactivation of antimicrobial drugs following the production of carbapenemases is one of the most frequent mechanisms of resistance [El-Gamal MI, Brahim I, Hisham N, Aladdin R, Mohammed H, Bahaaeldin A. Recent updates of carbapenem antibiotics. Eur J Med Chem. 2017 May 5;131:185-195. doi: 10.1016/j.ejmech.2017.03.022. Epub 2017 Mar 16; European Centre for Disease Prevention and Control. Carbapenem-resistant Enterobacteriaceae, second update – 26 September 2019. ECDC: Stockholm; 2019. Available at: https://www.ecdc.europa.eu/en/publications-data/carbapenem-resistant-enterobacteriaceae-second-update]”.

R2:” Materials and Methods: Line 77-82. The authors should re-word this part with shorter sentences. This part is confusing.

AA: We thank the Reviewer for this suggestion.

“…For the purposes of the present study only the first positive rectal swab was included. Colonization was defined as gastrointestinal tract carriage following the detection of CRE form rectal swab or feces, whereas infection definition was the same used by the National Healthcare Safety Network (NHSN) protocols [National Healthcare Safety Network (NHSN) Patient Safety Component Manual...” The text was modified accordingly.

R2:” Results: All the collected data are shown in tables. Although the tables are informative, the authors could use different types of graphs to plot the data (for example the cake charts might help the reader to follow the obtained results).

AA: We thank the Reviewer for this important suggestion. A graph on the distribution of patients in the wards of admission was added.

R2:” Results: The authors should clarify the meaning of the acronyms ICU and CPE”

AA: We thank the Reviewer for his/her recommendation. As previously highlighted, the acronym ICU was described in the abstract and CPE refers to carbapenemase producing Enterobacteriaceae. The manuscript was edited accordingly.

R2:” Results: At the beginning of the sentence is better to write the word instead to refer to the number to make reading smoother (for example line 231 and many other)”

AA: We thank the Reviewer for this comment. The text was revised and edited accordingly.

Reviewer 3 Report

In this manuscript authors present an extremely drug resistant strain of Klebsiella pneumoniae and its association with class A and D carbapenemases in Italy. This could have been an important XDR gene association study however data presented in the manuscript is not enough to prove the claims.

Contrary to authors claim, following articles have reported that K. pneumoniae drug resistance has an association with class A and D carbapenemases, two are from Italy

https://www.ncbi.nlm.nih.gov/pmc/articles/PMC7508593/

https://www.ncbi.nlm.nih.gov/pmc/articles/PMC7266126/

https://www.frontiersin.org/articles/10.3389/fmicb.2019.02767/full

Having said that this study still provides an important piece of research literature if following additions are made,

  • PFGE/MLST analysis data and method should be provided in the manuscript.
  • A robust statistical analysis of qPCR data to prove the association between drug resistance and genes (KPC & OXA48) should be performed.

Minor,

Role of age should be discussed in the demographic section.

Table 2 can be presented differentiating patients with one or more comorbidities.

Author Response

Reviewer #3:

R3:”In this manuscript authors present an extremely drug resistant strain of Klebsiella pneumoniae and its association with class A and D carbapenemases in Italy. This could have been an important XDR gene association study however data presented in the manuscript is not enough to prove the claims.”

AA: We thank the Reviewer for his/her comments and the contribution.

R3: “Contrary to authors claim, following articles have reported that K. pneumoniae drug resistance has an association with class A and D carbapenemases, two are from Italy:

https://www.ncbi.nlm.nih.gov/pmc/articles/PMC7508593/

https://www.ncbi.nlm.nih.gov/pmc/articles/PMC7266126/

https://www.frontiersin.org/articles/10.3389/fmicb.2019.02767/full”

AA: We thank the Reviewer for having highlighted this important point. Multiple carbapenemase production was reported worldwide; however, we referred to the first detection of KPC and OXA-48 carbapenemases. Following his/her important suggestion, we modified the sentence as follows: “…this is the first report of K. pneumoniae co-harbouring KPC and OXA-48 carbapenemases in Italy.”

R3:”Having said that this study still provides an important piece of research literature if following additions are made:

  • PFGE/MLST analysis data and method should be provided in the manuscript.”

AA: We thank the Reviewer for this request of clarification. Dendrogram and electrophoretic bands of K.pneumoniae strains, carried out by PFGE analyses, were submitted as supplementary material. A more detailed description of PFGE analysis was added in the “Materials and Methods” section, as follows: “Bacteria were suspended in agarose disk before DNA extraction and purification. Bacterial genomes were cut by ApaI restriction enzyme and single fragments were separated by agarose PFGE using a Clamped Homogeneous Electric Fields DRII System (BIORAD). Image of gel was captured by Image Master Program and position of electrophoretic bands and the phylogenetic dendrogram were obtained by GelCompar II (Applied Math).” Details of MLST analyses were added in the supplementary material.

  • ”A robust statistical analysis of qPCR data to prove the association between drug resistance and genes (KPC & OXA48) should be performed.”

AA: We thank the Reviewer for having highlighted this point. Unfortunately, we could not address this request because of the methodology. Following the current legislation [D. Lgs. n. 332/2000, based on European recommendation 98/79/CE and UE 2017/746], only kit marked as CE-IVD can be employed. On this basis, we adopted the commercial kit Allplex EnteroDR assay (Seegene) for screening. Allplex Entero DR assay is an accurate and fast method for the detection of the most prevalent antibiotic resistance genes. However, the protocol of analysis on bacterial colonies did not allow to quantify gene expression, consequently, only a qualitative analyses could be performed More details on the methodology are available at the following link: www.arrowdiagnostics.it/download/microbiologia/resistenzeadantimicrobici/BROCHURE_AllplexEntero-DRAssay.pdf. We will plan further analyses in the near future.

R3:”Role of age should be discussed in the demographic section.”

AA: We thank the Reviewer for this recommendation. The regression analysis did not prove a role of age in the occurrence of the planned outcomes. However, the findings could be affected by the poor sample size. Then, an ad hoc sentence was added in the manuscript (Please, see discussion section, page 8).

R3:”Table 2 can be presented differentiating patients with one or more comorbidities.”

AA: We thank the Reviewer for his/her suggestion. A paragraph on diabetes was added in the Results and Discussion sections to highlight the significant association with mortality. Moreover, the table no. 2 was edited accordingly, adding on the number of single and multiple comorbidities.

Round 2

Reviewer 1 Report

Thanks

Reviewer 3 Report

Authors have addressed all the concerns and comments. I recommend the manuscript for publication